# Anti-Inflammatory Effects of Ang-(1-7) Bone-Targeting Conjugate in an Adjuvant-Induced Arthritis Rat Model

**DOI:** 10.3390/ph15091157

**Published:** 2022-09-17

**Authors:** Sana Khajeh pour, Arina Ranjit, Emma L. Summerill, Ali Aghazadeh-Habashi

**Affiliations:** 1College of Pharmacy, Idaho State University, Pocatello, ID 83209, USA; 2College of Health, Idaho State University, Pocatello, ID 83209, USA

**Keywords:** rheumatoid arthritis, inflammation, renin-angiotensin system, Angiotensin-(1-7)

## Abstract

Rheumatoid arthritis (RA) is a chronic inflammatory condition of synovial joints that causes disability and systemic complications. Ang-(1-7), one of the main peptides in the renin-angiotensin (Ang) system (RAS), imposes its protective effects through Mas receptor (MasR) signaling. It has a short half-life, limiting its feasibility as a therapeutic agent. In this study, we evaluated the anti-inflammatory effects of Ang-(1-7)’s novel and stable conjugate (Ang. Conj.) by utilizing its affinity for bone through bisphosphonate (BP) moiety in an adjuvant-induced arthritis (AIA) rat model. The rats received subcutaneous injections of vehicle, plain Ang-(1-7), or an equivalent dose of Ang. Conj. The rats’ body weights, paws, and joints’ diameters were measured thrice weekly. After 14 days, the rats were euthanized, and the blood and tissue samples were harvested for further analysis of nitric oxide (NO) and RAS components’ gene and protein expression. The administration of Ang. Conj. reduced body weight loss, joint edema, and serum NO. Moreover, the Ang. Conj. treatment significantly reduced the classical arm components at peptide, enzyme, and receptor levels while augmenting them for the protective arm. The results of this study introduce a novel class of bone-targeting natural peptides for RA caused by an inflammation-induced imbalance in the activated RAS. Our results indicate that extending the half-life of Ang-(1-7) augments the RAS protective arm and exerts enhanced therapeutic effects in the AIA model in rats.

## 1. Introduction

Rheumatoid arthritis (RA) is a chronic and inflammatory autoimmune disorder affecting about 1% of the world population [1]. It is a chronically inflamed condition of synovial joints, eventually progressing to disability and systemic complexities leading to early fatality [2]. RA affects large and small joints through symmetrical polyarthritis. This disorder typically presents between the ages of 30 to 50 and is the most common inflammatory arthritis. The etiology of RA has not been fully defined, but genetic and environmental factors are considered the main initiators of the disease. Complications associated with RA include increased risk of developing cardiovascular disease, cerebrovascular and coronary artery atherosclerosis [3], and abnormal skeletal health [4]. Joint destruction and systemic complications can be mediated by complex interactions among multiple immune cell types, growth factors, and cytokines [5]. Multiple systems orchestrate the complex and dynamic situation of inflammation in the body, including the renin-angiotensin system (RAS) [6]. One of the factors responsible for RA’s pathology and tissue damage is Ang II. Ang II can activate proinflammatory cytokines and initiate reactive oxygen species (ROS) production in vascular smooth muscle cells, neutrophils, and osteoclasts [7]. RA is associated with excess morbidity and mortality from cardiovascular complications [8,9]. Evidence suggests that common proinflammatory cytokines are involved in the development and progression of both atherosclerosis and cardiovascular complications [3,10]. Current therapeutic options, such as analgesics, disease-modifying antirheumatic drugs, non-steroidal anti-inflammatory drugs, and biologics, are associated with severe and broad-spectrum side effects. There is a dire need for safer and more effective therapeutic options for RA management.

The RAS consists of proinflammatory and anti-inflammatory axes balanced in the normal physiological condition (Figure 1). As one of the RAS main vasoactive peptides, Ang II is produced by Ang-Converting Enzyme (ACE) from Ang I peptide. It mediates proinflammatory actions by binding to the Ang II type 1 receptor (AT1R) [11]. Whereas the protective RAS axis component, Ang-(1-7), is produced by the action of ACE2. It binds to the G-protein coupled MasR and exerts anti-inflammatory and anti-pressor effects [12]. The ACE/Ang II/AT1R arm overactivation in some pathological conditions switches the balance off towards the inflammatory axis. It is justifiable to endogenously increase the Ang-(1-7) level and augment the protective arm to suppress the overactivation of the proinflammatory axis and restore the disturbed balance.

Ang-(1-7) as a vasoactive peptide induces vasodilatory, anti-inflammatory, antifibrotic, antiangiogenic, and antihypertensive effects of MasR signaling. These beneficial actions of Ang-(1-7) make this peptide an attractive target for cardioprotective therapies. These protective effects have been demonstrated in numerous animal models of human diseases, including hypertension, diabetes mellitus, and atherosclerosis [13]. The administration of Ang-(1-7) is therapeutically effective in the experimental animal model of RA due to the activity mediated by the reduction of neutrophil accumulation, inhibition of cytokine release, and improvement of joint hyper nociception [12,14]. Additionally, Ang-(1-7) exhibits anti-inflammatory behavior by reducing cytokine release, tissue damage, leukocyte attraction, and fibrosis [7].

Activating the RAS induces inflammation by signaling AT1R by Ang II on leukocytes. After RA’s initiation and development, inflammatory cells infiltrate the articular synovial tissues and consequently secrete inflammatory cytokines, such as TNF-α, IL-1, and IL-6 [15]. In rodent models of AIA through augmentation of the RAS protective arm by Ang-(1-7) or the Mas agonist AVE 0991, the AIA-induced neutrophil accumulation and production of proinflammatory mediators such as TNF-a, IL-1β, and CXCL1 in prearticular tissue were diminished, and paw histopathological markers were significantly reduced [12]. Ang-(1-7) also improved paw histological changes and normalized cytokine biomarkers in the Collagen-induced arthritis rat model [16]. Furthermore, increased levels of NO in plasma [17,18], articular [19] fluid, and a high expression of iNOS in hyperplastic synovium [17,20] and chondrocytes [21] have been reported in different models of AIA. This elevation of NO and ROS are related to the presence of macrophages in the inflamed tissues due to higher expression of iNOS induced by inflammatory cytokines. In a study using cultured human aortic smooth muscle cells (HASMC), the expression of inducible nitric oxide synthase (iNOS) and the release of nitric oxide (NO) were stimulated by both Ang II and IL-1β. Treatment with Ang-(1-7) inhibits the NO production, which Mas receptor antagonists block such effect [22]. These findings indicate that the Mas receptor activation by Ang-(1-7) prevents neutrophil influx, cytokine production, and No release and, as a result, significantly improves arthritis in rats and mice AIA models. However, Ang-(1-7) has an extremely short half-life (3–15 min) due to the rapid systemic clearance, which restricts its potential therapeutic benefits [23]. Therefore, introducing an appropriate drug delivery system that extends the systemic half-life of Ang-(1-7) could offer a most-needed safe [24] and effective therapeutic option for RA. As mentioned above, Ang-(1-7) presented protective effects by reducing proinflammatory cytokine levels in the plasma, HASMC, and joint tissues. In the current study, we further investigated the impact of inflammation on the RAS at the cellular level and focused on exploring the cardiovascular complication of RA. We additionally studied the protective effects of Ang. Conj. treatment, as a stable form of Ang-(1-7), on the RAS in the enzyme, peptide, and receptor levels of the heart, kidney, liver, and lung tissues.

We designed a bone-targeting peptide delivery system that actively targets the bone, makes a drug depot, slowly releases the active peptide, and prolongs its circulation half-life [25]. This approach seems promising in the delivery strategy for treating bone disorders like RA, osteoporosis, and cancers with bone complications [26]. In this targeted drug delivery, a drug is conjugated with a bone targeting moiety like BP with linkers like PEG. The BP has an intrinsic affinity to the hydroxyapatite of the bone, which serves as a drug reservoir and increases its metabolic stability [27]. Similar peptides such as parathyroid hormone [28], salmon calcitonin [29], and osteoprotegerin [30] showed higher therapeutic efficacy than their parent analog when delivered through a bone-targeting approach due to the half-life extension of conjugated peptide drugs. It is worth mentioning that contrary to the nitrogen-containing bisphosphonates, the BP moiety of the Ang. Conj. does not possess any significant pharmacological effect [28].

Using a gamma counter radioassay, we have shown that Ang. Conj. has a longer half-life [25]. The radioassay methods have some limitations in their application for the quantitative measurement of radiolabeled material. We are currently developing an LC-MS/MS method to quantify Ang. Conj. in plasma to confirm our previous PK results. Assuming a direct relationship between the plasma concentration of Ang-(1-7) and its anti-inflammatory effect, we investigated Ang. Conj. therapeutic effects using a rat model of RA compared to plain Ang-(1-7) and elucidated the impact of bone-targeting delivery and stability improvement strategy on enhancing their anti-inflammatory effects. To make the comparison more effective and the effect size more significant, we increased the dosing interval from the previously used daily regimen [16] to 3×/week.

In this study, we focused on investigating the cardiovascular complication of RA. We aimed to explore the impact of the inflammation on the RAS in the heart, kidney, liver, and lung tissues and investigate the protective effects of Ang. Conj. treatment. We did not include the paw tissue or inflammatory cytokines measurement, as it was previously shown by Liu et al. that Ang-(1-7) improves paw histological and normalized cytokine biomarkers changed in the Collagen-induced arthritis rat model [16]. Increased plasma and synovial tissue NO levels due to high expression of iNOS have been reported in patients with RA [31,32]. In the rat model of AIA, systemic administration of selective and nonselective iNOS halted the development of the disease [17,33]. Similarly, we evaluated Ang. Conj. systemic anti-inflammatory effects through assessment of serum concentration of NO.

## 2. Results

### 2.1. Ang. Conj. Treatment Improved Body Weight Gain and Reduced Paw and Joint Swelling

Adjuvant-induced Arthritis (AIA) emerged 8–10 days post-adjuvant injection. It manifested itself by redness of the paw and erythema of ankle joints, followed by the involvement of the metatarsal and interphalangeal joints. The symptoms spread progressively with time into other parts of the hind and forepaws. The weight, paw, and joint measurements were done thrice per week. The paw and joint diameters are reported as percentage change on day 24 compared to day 0.

The percentage change of animals’ body weight gain was significantly reduced in AIA animals and treatment with Ang-(1-7) and Ang. Conj. restored the body weight gain over time (Figure 2A). The absolute mean ± SEM values of body weight (g) on day 24 were 409.6 ± 8.4 for control, 290.0 ± 2.8 for inflamed, 362.0 ± 19.4 for Ang-(1-7), and 373 ± 16.6 for Ang. Conj. groups. The weight gain percentage value in the inflamed (12.6 ± 5.5%) and Ang-(1-7)-treated group (31.6 ± 5.3%) was significantly lower than the control group (50.1 ± 2.5%). However, the value in the Ang. Conj-treated group (34.2 ± 4.5%) was comparable to that of the control group (Figure 2A). As a measure of signs and symptoms of arthritis, the arthritis index (AI) was significantly higher in the inflamed group indicating efficacious arthritis induction. The therapeutic efficacy of Ang-(1-7) and Ang. Conj. was noticeable after administration of three consecutive doses (~6 days), but it was statistically significant 10 days after treatment started (Figure 2A,B). The right and left joints’ diameter percent changes at the end of the experiment compared to day 0 in the non-treated inflamed rats (29.0 ± 9.0% and 27.7 ± 7.4%) were significantly higher than in the control group (2.7 ± 1.2 and 3.8 ± 0.8%). The drug treatment impacted the right and left joints swelling as it was substantially lower in Ang. Conj. (3.6 ± 1.6% and 2.5 ± 1.4%) and Ang-(1-7) (9.6 ± 2.5% and 12.4 ± 7.4%) groups (Table 1 and Figure 2C,E). The absolute mean ± SEM values for the right joint diameter (mm) on day 24 were 6.7 ± 0.06, 8.6 ± 0.06, 7.2 ± 0.06, and 6.7 ± 0.08, and for the left joint diameter (mm) were 6.8 ± 0.15, 8.6 ± 0.03, 7.1 ± 0.08, and 6.8 ± 0.08 for the control, inflamed, Ang-(1-7)-treated, and Ang. Conj.-treated groups, respectively. The same trend was seen for the left and right paw diameter percent changes but did not reach a significant difference due to high variability (Table 1, Figure 2D,F). The absolute mean ± SEM values on day 24 for the right paw diameter (mm) were 4.2 ± 0.05, 5.3 ± 0.08, 4.6 ± 0.03, and 4.2 ± 0.06, and for the left paw diameter (mm) were 4.3 ± 0.04, 5.1 ± 0.04, 4.4 ± 0.02, and 4.3 ± 0.02 for the control, inflamed, Ang-(1-7)-treated, and Ang. Conj.-treated groups, respectively.

### 2.2. The Arthritis-Induced High Serum Nitrate and Nitrite Levels Were Reduced after Treatment with Ang-(1-7) and Ang. Conj.

The serum nitric oxide (NO) level was significantly elevated in inflamed animals (5.15 ± 1.02 ng/mL) compared to healthy control rats (0.22 ± 0.13 ng/mL). Ang. Conj. treatment significantly reduced the serum NO concentration to 0.95 ± 0.52 ng/mL, which was comparable to the control group. Although Ang-(1-7) treatment reduced the serum NO concentration to 2.35 ± 0.73 ng/mL, this reduction did not happen to the same extent as observed after treatment with Ang. Conj. The serum NO level in Ang-(1-7)-treated rats was not significantly different from the inflamed or control group (Figure 3).

### 2.3. Treatment with Ang-(1-7) or Ang. Conj. Increased Ang-(1-7) and Reduced Ang II Peptides Levels in Plasma

The plasma level of Ang-(1-7) and Ang II peptides and their ratio are presented in Figure 4. These data confirmed the treatment of AIA rats with Ang. Conj. significantly increases the Ang-(1-7) plasma levels (1.45 ± 0.22 ng/mL) compared with the control (0.75 ± 0.08 ng/mL), inflamed (0.19 ± 0.05 ng/mL), and Ang-(1-7)-treatment (0.90 ± 0.17 ng/mL) groups (Figure 4A). The Ang II plasma concentrations, on the other hand, present a reverse trend. The AIA significantly elevates Ang II plasma levels in the inflamed group (1.70 ± 0.37 ng/mL), which reduced to a comparable plasma concentration of the control group (0.27 ± 0.02 ng/mL) after treatment with Ang-(1-7) (0.30 ± 0.036 ng/mL) or Ang. Conj. (0.21 ± 0.02 ng/mL) groups (Figure 4B).

The Ang-(1-7)/Ang II ratio was significantly higher in the Ang. Conj-treated group (6.30 ± 1.89) compared with the healthy-control (4.10 ± 1.42), inflamed (0.1 ± 0.03), and Ang-(1-7) (2.00 ± 0.60) groups (Figure 4C).

### 2.4. Treatment with Ang-(1-7) or Ang. Conj. Reversed the AIA-Induced Changes in ACE1, ACE2, MasR, and AT1R Gene Expression in Different Tissues

The mRNA expression of ACE1, ACE2, AT1R, and MasR in the heart, lung, liver, and kidney, are shown in Figure 5. ACE1 and AT1R gene expression levels significantly increased in all tested tissues of the inflamed rats, and ACE2 gene expression was reduced in the heart, lung, liver, and kidney. MasR gene expression increased in the heart and decreased in the lung and kidney, with no change observed in the liver. Treatment with Ang-(1-7) and Ang. Conj. (in a higher magnitude) significantly reversed all changes. However, in the heart tissue, the increased MasR expression due to AIA was further increased by Ang. Conj. ACE2/ACE1 and MasR/AT1R gene expression ratios in all tested tissues were reduced due to AIA, and treatment with Ang-(1-7) normalized them while Ang. Conj. increased them several folds.

### 2.5. Treatment with Ang-(1-7) or Ang. Conj. Reversed the AIA-Induced Changes in ACE1, ACE2, MasR, and AT1R Protein Expression in Different Tissues

Data shown in Figure 6 represent the significant changes in relative protein density of ACE1, ACE2, MasR, and AT1R in the heart, lung, liver, and kidney tissues due to AIA. These changes were normalized by Ang-(1-7) or Ang. Conj. treatment and the effects were more pronounced in the case of the latter. The individual proteins WB results indicate (i) AIA caused a significant increase in the ACE1 protein expression in the heart, liver, and kidney tissues of the inflamed rats and Ang-(1-7) or Ang. Conj. treatment reversed it, which was more efficient in the latter case, and brought it back to the control group level. (ii) AIA significantly reduced the ACE2 expression in the heart and lung tissues, and treatment with Ang-(1-7) or Ang. Conj. normalized it. In the case of the liver and kidney, AIA resulted in a similar change trend but was not significant. (iii) MasR’s expression was significantly reduced in inflamed animals’ lung, liver, and kidney tissues, which were again normalized by Ang-(1-7) or Ang. Conj. treatments. (iv) The AT1R expression was significantly increased in all tissues other than the kidney in the inflamed group and Ang. Conj. treatment reversed the expression more efficiently than Ang-(1-7). (v) ACE2/ACE1 and MasR/AT1R protein expression ratios in all tested tissues were reduced by AIA, and treatment with Ang-(1-7) or Ang. Conj. normalized them.

## 3. Discussion

The administration of Ang-(1-7) or Ang. Conj. impacted the activated RAS, which was more pronounced in the Ang. Conj. case. The anti-inflammatory effects of Ang-(1-7) have been reported previously [12]. However, its short half-life hampers such an application’s feasibility. The findings of this study, for the first time, indicate that it is feasible. We designed a bone-targeting drug conjugate of Ang-(1-7); Ang. Conj. and improved its pharmacokinetics, leading to an enhancement of pharmacodynamic properties [25].

The results of the present study demonstrate an augmentation of the RAS protective arm by exogenous administration of Ang-(1-7) as a stable conjugate, Ang. Conj., in an AIA model of RA. Such a boost exerts significant anti-inflammatory effects by restoring weight gain loss, reducing the paw and joint swelling, and diminishing the increased NO level and AI due to inflammation (Figure 2 and Figure 3). The anti-inflammatory effect of Ang. Conj. reducing and normalizing the NO serum concentration followed the same trend of change in Ang-(1-7) and Ang II plasma levels. Although the NO serum levels in Ang-(1-7) or Ang Conj.-treated groups did not reduce to the exact same level as the control group; there was no significant difference between their NO values. Similarly, the therapeutic efficacy of Ang-(1-7) and Ang. Conj. on reducing AI was not significantly noticeable until ten days after treatment started (Figure 2A,B). This observation could be attributed to the time required for Ang-(1-7) plasma concentration to release from the conjugate and to reach an adequate steady-state concentration. The two weeks treatment period with a low dose of Ang-(1-7) or Ang. Conj. in this pilot study presents a promising positive outcome, which can be potentiated by increasing the dose and duration. These findings are mostly in line with reestablishing the disturbed balance between the classical and protective arms, which is presented as ratios of Ang-(1-7)/Ang II peptides (Figure 4), or ACE2/ACE1 and MasR/AT1R at the gene or protein levels (Figure 5 and Figure 6).

In agreement with previous reports [34], the results of this study (Figure 4, Figure 5 and Figure 6) indicate that induced inflammation in the AIA rats alters the balance of the RAS components in plasma peptides and enzyme and receptor levels in all tested tissues. The ACE and ACE2 enzyme expression alteration affect the plasma Ang-(1-7), Ang II, and their ratio (Figure 4). This observation indicates that the lower expression of ACE2 resulted in a lower plasma concentration of the vasodilator peptide, Ang-(1-7). This effect was reported in the heart and kidney tissues of the AIA rats and most likely applies to other tissues as well [34]. Considering the association of RAS with cardiovascular diseases, the observed RAS imbalance could be attributed to the well-known effect of inflammation in increasing cardiovascular risks [35]. It is notable that Ang II promotes proinflammatory outcomes and is elevated in many cardiovascular conditions, such as hypertension, atherosclerosis, and coronary heart disease, by stimulating the production of different inflammatory mediators and their migration into sites of tissue injury [36,37]. Consistent with the proinflammatory actions of Ang II, treatment with ACE inhibitors (ACEIs) and AT1R blockers (ARBs) diminishes the production and release of inflammatory mediators in models of inflammation [38,39,40,41,42,43]. ACE2, by degrading the Ang II to Ang-(1-7) and MasR signaling, counters the +Ang II proinflammatory effects [44,45]. There were no significant changes in the gene expression of MasR in any tested tissues due to AIA, but it was significantly increased in the case of AT1R in the heart and liver tissues (Figure 5). These gene expression changes variably translated to changes in protein expression as the lung, liver, and kidney tissues were presented with lower MasR receptor density and all other than the kidney tissue had higher AT1R expression.

Nevertheless, the MasR/AT1R protein ratio was lower in all tissues in non-treated AIA animals. This observation exerts a further change in balance in the activated RAS components toward the classical arm deleterious effects. The observed significant elevation of AT1R and reduced MasR/AT1R (Figure 6) support the notion that arthritis can change the vasodilation–vasoconstriction balance as there will be less target protein, MasR, for coupling with already reduced vasodilator Ang-(1-7), and, on the other hand, more AT1R is available to be activated by a higher concentration of a vasoconstrictor, Ang II. We only measured the Ang peptides levels in plasma, which were significantly altered in AIA rats (Figure 4); however, considering the lowered ratio of the ACE/ACE expressions, it most likely is the case in all tissue (Figure 6).

The RAS exerts its physiological effects through AT1R, AT2R, and Mas receptors. Ang II has an affinity for AT1R and AT2R (with offsetting responses), but most Ang II effects are mediated through AT1R [46]. In the present study, we focused only on the AT1R and MasR; however, the AT2R receptor gene and protein expressions can also be impacted by inflammation and should be considered when the results are interpreted.

The depicted results in Figure 4, Figure 5 and Figure 6 imply that treating AIA animals with plain Ang-(1-7) could compensate for the harmful impacts of activated RAS and its balance shift toward the classical arm. The restoration of the balance and exertion of anti-inflammatory effects was more significant and pronounced when Ang. Conj. was used. The improved efficacy can be attributed to the intermittently prolonged effect of Ang-(1-7) through MasR signaling. These observations are in concert with a previous study on the activation of MasR using its agonist, AVE 0991, in experimental models of arthritis [12]. The Ang. Conj. thrice-weekly application for two weeks significantly increased the reduced plasma Ang-(1-7), reduced the increased plasma level of Ang II, and restored the healthy control ratio of those peptides (Figure 4). The observed alteration of Ang peptides was in line with ACE and ACE2 gene and protein expression, which resulted in an overall significant improvement of the ACE2/ACE ratio in all tissues. The MasR/AT1R presents a similar trend in all tissues, although their expressions do not precisely follow suit. These findings indicate that the administration of Ang. Conj. is similar to treatment with ACEIs and ARBs and diminishes the anti-inflammatory effects of Ang II in different tissues to manage arthritis, renal, cardiovascular, pulmonary, and hepatic disease. For confirmation of these results, a head-to-head comparison study is warranted.

## 4. Materials and Methods

### 4.1. Animals

Adult male Sprague-Dawley rats weighing 200–250 g were obtained from Charles River (Wilmington, MA, USA) and were housed under ambient temperature and ventilation with 12-h day and night cycles. Rats were kept in standard cages with free access to drinking water and regular rat chow ad libitum. After 72 h of acclimatization, rats were randomly divided into healthy-control (*n* = 6), inflamed (*n* = 6), Ang-(1-7) (*n* = 5), and Ang. Conj. (*n* = 5) groups. The study protocol was approved by the Animal Care Facility Committee of Idaho State University (Protocol #772, 22 September 2021).

### 4.2. Arthritis Induction in Rats

AIA is associated with pain and discomfort for animals. To address that issue, we used a previously reported arthritis induction method that avoids unnecessary pain and discomfort while the inflammation effects are considerably noticeable [47,48]. This model is proven suitable for pharmacokinetics and pharmacodynamics studies. For induction of arthritis, animals were injected at the tail base on day 0 with a single dose of 200 µL of 50 mg/mL of Mycobacterium *butyricum* in squalene (Difco Laboratories, Detroit, MI, USA) (all groups except the healthy-control) or pyrogen-free sterile saline (healthy-control) to induce adjuvant arthritis. Subsequently, rats were monitored daily and assessed for the emergence of arthritis by assigning an arthritis index score for each rat. Arthritis index is a macroscopic scoring system [48]: for each hind paw on a 0–4 scale, 0 = no sign; 1 = single joint involved; 2 = more than one joint and ankle involved; 3 = several joints and ankle involved with moderate swelling; 4 = involvement of several joints and ankle with severe swelling. For each forepaw on a 0–3 scale, 0 = no sign; 1 = single joint involved; 2 = more than one joint and wrist affected; 3 = involvement of wrist and joints with moderate-to-severe swelling. The index was calculated by adding all the above scores to attain a maximum of 14. An arthritis index score of ≥5 was considered infliction of the disease, and its early signs and symptoms were evident typically in 8–10 days after adjuvant injection.

### 4.3. Body Weight, Paw, and Joint Diameter Measurement

Paw and joint diameters are indicatives of edema in rats and were measured three times per week, always at the same hour, using a micrometer caliper (Mitutoyo Canada Inc., Toronto, ON, Canada). The change in the rats’ body weights was measured and recorded every other day using the animal balance.

### 4.4. Animal Dosage Regimens, Treatment, and Sampling

Ang-(1-7) (0.6 mg/kg) or Ang. Conj., containing an equivalent dose of Ang-(1-7), was dissolved in sterile normal saline and administered subcutaneously thrice per week for 14 days after the emergence of the signs and symptoms of inflammation at least in one hind on 8–10 days post adjuvant injection. The healthy-control and inflamed groups were injected with drug-free normal saline. At the end of the experiment, rats were anesthetized with isoflurane/oxygen, and blood samples were collected by cardiac puncture. Subsequently, the heart, kidney, lungs, and liver tissues were rapidly removed and washed with saline. Serum was separated from a portion of the collected blood that was not treated with an anti-coagulant for serum NO concentration assay. Fifty microliters of protease inhibitor cocktail was added per each 1 mL of blood. The cocktail was composed of 1 mM of p-hydroxymercury benzoate, 30 mM of 10-phenanthroline, 1 mM of phenylmethylsulfonyl fluoride, 1 mM of the pepstatin-A enzyme, and 7.5% of ethylenediaminetetraacetic acid. After centrifugation for 10 min at 2500× *g*, the plasma was separated. All samples were stored at −80 °C until further analysis.

### 4.5. Measurement of the Serum NO Concentration

According to the manufacturer’s protocol, serum NO concentration was quantified using a Nitrate/Nitrite Colorimetric Assay Kit (#BCCB4059; Sigma, St. Luis, MO, USA). Briefly, blood samples were thawed, and each kit component was allowed to come to room temperature. Then, in a 96-well plate, 0, 20, 40, and 80 µL of 100 µM nitrite standard solution and 70 μL of each serum sample were added in triplicates. To measure the total nitrate and nitrite concentration, 10 µL of nitrate reductase and enzyme cofactor were added to the serum samples and shaken for 2 h. Then Griess reagents A and B were added to each well and incubated for 5 and 10 min, respectively. The absorbance intensity as an indicator of the sample NO concentration was measured at 540 nm in a microplate reader.

### 4.6. Ang Peptides Extraction and Quantification Using Liquid-Chromatography in Tandem with Mass Spectrometry (LC-MS/MS)

Ang peptides were extracted from plasma using solid-phase extraction (SPE) based on a previously published method with minor modifications [49]. Briefly, to 200 μL of plasma samples, 50 μL of the [Asn^1^, Val^5^] Ang II (IS) (100 ng/mL) was added and acidified with formic acid (final concentration of 0.5%). After a brief vortex mixing, the samples were loaded into preconditioned Waters C18 SPE cartridges (#WAT020805, Milford, MA, USA) with 2 mL of ethanol and water each by a wash step with 2 mL of deionized water. A positive nitrogen flow was applied to dry the cartridges. 2.5 mL of 5% formic acid in methanol was used to elute the Ang peptides. The eluted solution was collected and dried under the stream of nitrogen. The dried samples were reconstituted in 100 µL of 0.1% formic acid in acetonitrile: water (16:84), and 20 µL of it was injected into the LC-MS/MS system.

The plasma level of Ang-(1-7) and Ang II peptides, as one of the critical biomarkers of the protective and classical arms of the RAS, was measured using a validated LC-MS/MS method [49] in multiple reaction mode (MRM). The system was composed of liquid chromatography (Shimadzu, Columbia, MD, USA) with a controller (CBM-20A), two binary pumps (LC-30AD), an autosampler (SIL-30AC), and an AB Sciex (Foster City, CA, USA) QTRAP 5500 quadrupole mass spectrometer in positive electrospray ionization mode (ESI). The chromatograms were monitored and integrated by the Analyst 1.7 software (AB Sciex, Foster City, CA, USA).

The LC separation was performed on an analytical reversed-phase column Kinetex^®^ 1.7 μm, C-18, 100 × 2.1 mm (Phenomenex, Torrance, CA, USA) by a combination of A: 0.1% formic acid in water and B: 0.1% formic acid in acetonitrile as mobile phases at a flow rate of 0.2 mL/min. The mobile phase gradient started at 5% B and increased to 30% B in 5 min, kept at 30% B for 5 min, returned to 5% B in 3 min, and held at 5% B for 2 min before the next injection for column re-equilibrium.

The positive electrospray ionization parameters were as follows: capillary voltage; 5.5 kV, temperature; 300 °C, declustering potential (DP); 100 V, and collision cell exit potential (CXP); 15 V. LC-MS/MS was performed with MRM transitions of *m*/*z* 300.6 → 371.2 (Ang-(1-7)), *m*/*z* 349.7 → 400.2 (Ang II), and *m*/*z* 516.5 → 769.4 (IS). Nitrogen was used as collision gas, and the collision energies were set at 20–30 eV. A calibration curve using peak height ratio (analyte over IS) was constructed over the concentration range of 500 pg/mL to 10 ng/mL in plasma and used to measure the Ang peptides’ levels in plasma samples.

### 4.7. Quantitative Polymerase Chain Reaction (qPCR)

Total RNA was extracted from fifty mg tissue samples using the Quick-RNA™ Miniprep Plus kit (Zymo Research, Irvine, CA, USA) according to the manufacturer’s protocol. Briefly, the samples were lysed into a yellow Spin-Away™ Filter in a collection tube and centrifuged to remove the genomic DNA. Then 95–100% ethanol was added to the flow-through (1:1) and mixed well. This mixture was transferred into a green Zymo-Spin™ IIICG Column in a collection tube and centrifuged. The flow-through was discarded. After DNase treatment, 400 μL of RNA Prep Buffer was added to the column, the solution was centrifuged, and the flow-through was discarded. This process was repeated with 700 μL of RNA Wash Buffer. To remove the wash buffer, 400 μL of RNA Wash Buffer was added, and the column was centrifuged for 1 min. The remaining solution was carefully transferred to a nuclease-free tube. A hundred μL of DNase/RNase-Free Water was added directly to the column matrix and centrifuged to elute the RNA.

Real-time qPCR mRNA analyses were performed using SYBR Green Supermix (Bio-Rad, Hercules, CA, USA). Relative expression of all genes was determined by the comparative threshold cycle method using 2^−ΔΔct^ normalized with GAPDH constitutive gene and expressed as fold change compared with control. All primers were designed based on the rat species, and a list of forward and reverse primers is shown in Table 2.

### 4.8. Western Blot

The relative density of the proteins of interest (ACE, ACE2, AT1R, and MasR) was determined in the heart, lung, kidney, and liver tissues according to a previously reported western blotting method [55]. One hundred mg of each thawed tissue was sectioned, mixed in 1.5 mL of RIPA buffer containing a complete mini protease inhibitor tablet (Sigma Aldrich, St. Louis, MO, USA), and mechanically homogenized. The samples were centrifuged, the supernatant was collected, and the protein concentrations were determined using a Qubit^®^ protein reagent (Thermo Fisher Scientific, Waltham, MA, USA) based on the manufacturer’s protocol. The same amount of protein was loaded onto each well and separated using an electrophoresis method by 4–12% tris-glycine gel. The proteins were then transferred to a polyvinylidene fluoride (PVDF) membrane, blocked in 5% skim milk in wash buffer, and incubated overnight at 4 °C with designated primary antibodies: ACE (ab25422; 1:1000), ACE2 (ab108252; 1:1000), MasR (ab66030; 1:1000), and AT1R (ab124734; 1:1000). As the housekeeping protein α-tubulin (ab 4074; 1:1000) was used. The next day, the membrane was washed in wash buffer containing 0.1% Tween 20 and incubated with secondary antibody (ab; 1:10,000) for 2 h at room temperature on a horizontal shaker. Visualization and density quantification of the images was carried out using Azure Biosystems Chemiluminescence Kit (190625-38; Azure Biosystems, Dublin, CA, USA). Results are presented as the ratio of densities of the band of interest over that of the housekeeping protein. The bands were quantified using ImageJ 1.53e (The National Institutes of Health and the Laboratory for Optical and Computational Instrumentation, LOCI, University of Wisconsin) software.

### 4.9. Statistical Analysis

Statistical analyses were performed by GraphPad Prism 8.0 statistical software (San Diego, CA, USA). Results are expressed as the mean ± SEM. One-way analysis of the variances (ANOVA) was used to evaluate the differences between groups after assessing the equality of means by the F-test, followed by Tukey multiple comparison post-hoc analysis. The level of significance was set at *p* < 0.05. In all cases, the *p*-value for F-test was lower than 0.05, except for right and left paw diameters.

## 5. Conclusions

In conclusion, the results of this study suggest that inflammation alters the balance of the RAS components, such as enzymes, peptides, and receptors, at the plasma and tissue levels. The exogenous administration of Ang-(1-7) and Ang. Conj. restores the imbalances in the activated RAS caused by inflammation. The observed superior Ang. Conj. protective effects in different tissue suggest that the bone-targeted delivery of Ang-(1-7) enhances its efficacy and can be a valuable therapeutic option for inflammatory diseases such as rheumatoid arthritis, renal, cardiovascular, pulmonary, and hepatic diseases.

## Figures and Tables

**Figure 1 pharmaceuticals-15-01157-f001:**
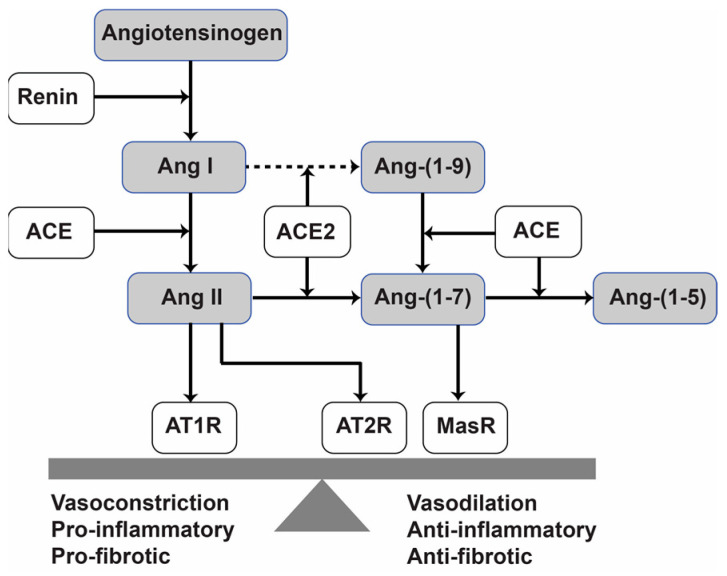
Schematic overview of the RAS. Ang—angiotensin, ACE—angiotensin-converting enzyme, ACE2—angiotensin-converting enzyme 2, MasR—Mas receptor, AT1R—angiotensin II type 1 receptor, AT2R—angiotensin II type 2 receptor.

**Figure 2 pharmaceuticals-15-01157-f002:**
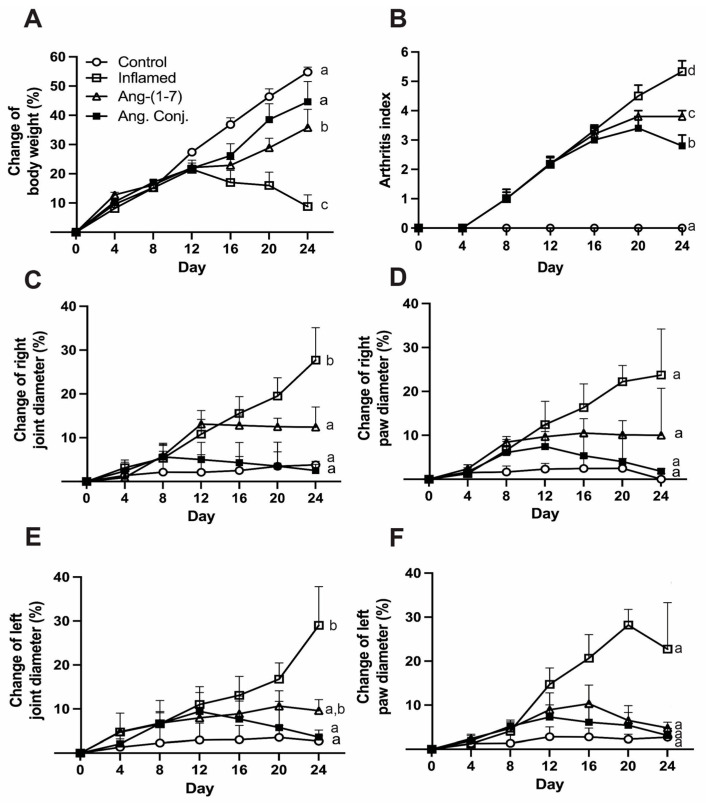
Effect of adjuvant-induced arthritis and treatment with Ang-(1-7) and Ang. Conj. on the percentage change of body weight (**A**) and the arthritis index values during the study period (**B**). The percentage change in the right joint (**C**), right paw (**D**), left joint (**E**), and left paw (**F**) diameters compared to day 0. The number of animals in control and inflamed groups were six and in Ang-(1-7) and Ang. Conj.-treated groups were five. The values are reported as mean ± SEM, and statistical analysis was done using one-way ANOVA with the Tukey multiple comparison post-hoc test. Groups labeled with different letters have significant differences between them (*p* < 0.05).

**Figure 3 pharmaceuticals-15-01157-f003:**
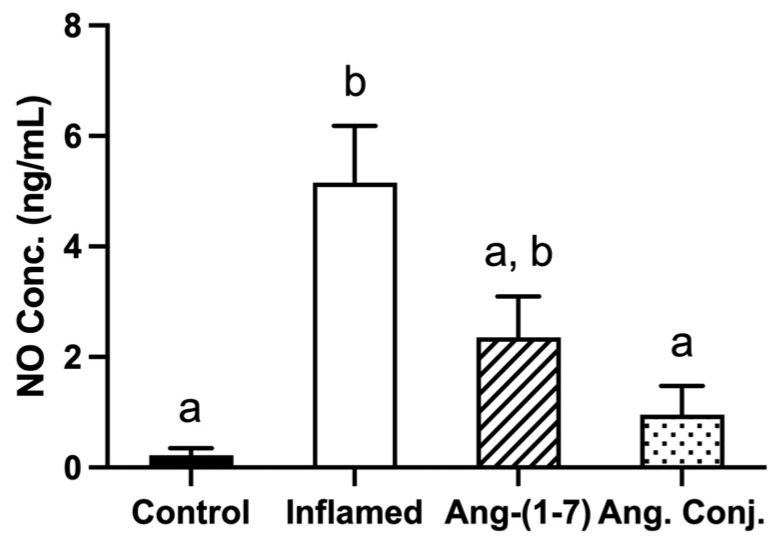
Effect of adjuvant-induced arthritis and Ang-(1-7) or Ang. Conj. treatment on serum NO concentrations at the end of the experiment. The number of animals in control and inflamed groups were six, and in Ang-(1-7) and Ang. Conj.-treated groups were five. The values are reported as mean ± SEM, and statistical analysis was done using one-way ANOVA with the Tukey multiple comparison post-hoc test. Groups labeled with different letters have significant differences between them (*p* < 0.05).

**Figure 4 pharmaceuticals-15-01157-f004:**
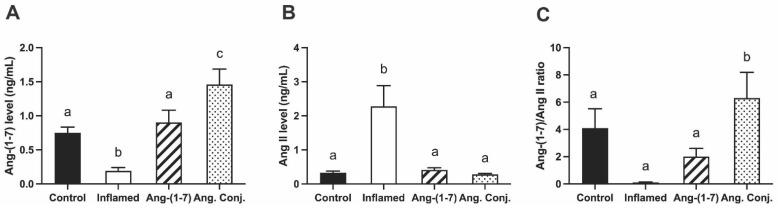
The plasma concentration of Ang-(1-7) (**A**), Ang II (**B**), and their ratio (**C**) in control (*n* = 6), inflamed (*n* = 6), Ang-(1-7)-treated (*n* = 5), and Ang. Conj.-treated (*n* = 5) groups. The values are reported as mean ± SEM, and statistical analysis was done using one-way ANOVA with the Tukey multiple comparison post-hoc test. Groups labeled with different letters have significant differences between them (*p* < 0.05).

**Figure 5 pharmaceuticals-15-01157-f005:**
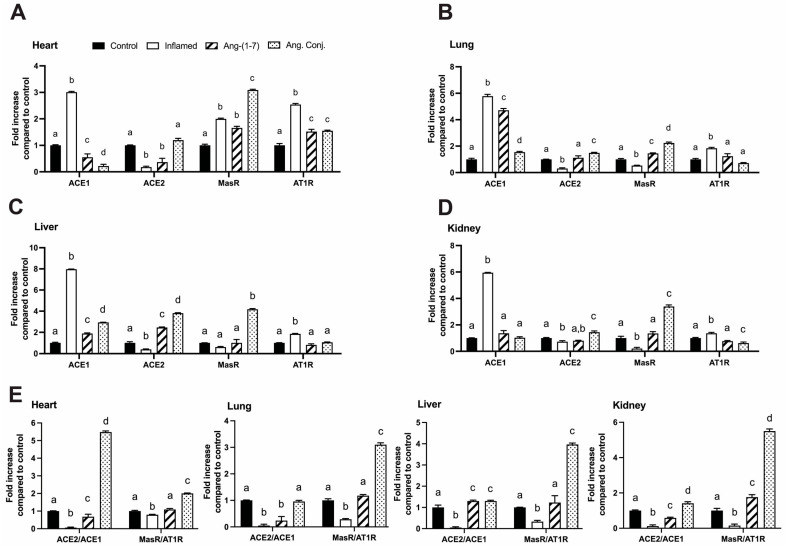
Gene expression of ACE1, ACE2, MasR, and AT1R levels in the heart (**A**), lung (**B**), liver (**C**), kidney (**D**) tissues, and the ratio of ACE2/ACE1 and MasR/AT1R in those tissues (**E**) in control (*n* = 6), inflamed (*n* = 6), Ang-(1-7)-treated (*n* = 5), and Ang. Conj.-treated (*n* = 5) groups. ACE1—angiotensin-converting enzyme 1, ACE2—angiotensin-converting enzyme 2, MasR—Mas receptor, AT1R—angiotensin II type 1 receptor. The values are reported as mean ± SEM, and statistical analysis was done using one-way ANOVA with the Tukey multiple comparison post-hoc test. Groups labeled with different letters have significant differences between them (*p* < 0.05).

**Figure 6 pharmaceuticals-15-01157-f006:**
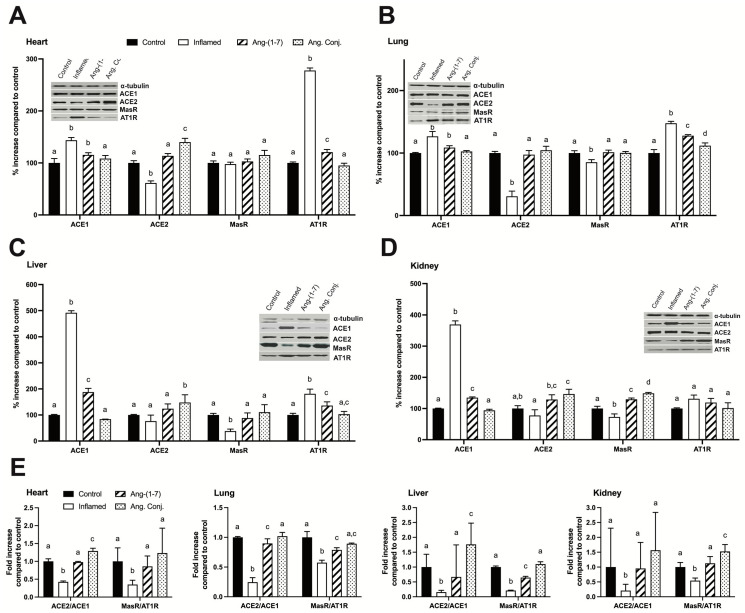
Protein expression of ACE1, ACE2, MasR, and AT1R, levels in the heart (**A**), lung (**B**), liver (**C**), kidney (**D**) tissues, and the ratio of ACE2/ACE1 and MasR/AT1R (**E**) in control (*n* = 6), inflamed (*n* = 6), Ang-(1-7)-treated (*n* = 5), and Ang. Conj.-treated (*n* = 5) groups. ACE1—angiotensin-converting enzyme 1, ACE2—angiotensin-converting enzyme 2, MasR—Mas receptor, AT1R—angiotensin II type 1 receptor. The values are reported as mean ± SEM, and statistical analysis was done using one-way ANOVA with the Tukey multiple comparison post-hoc test. Groups labeled with different letters have significant differences between them (*p* < 0.05).

**Table 1 pharmaceuticals-15-01157-t001:** The percentage change in paw and joint diameters in different treatment groups at the end of the experiment compared to day 0.

	Paw Diameter Percentage Change Mean (SEM)		Joint Diameter Percentage Change Mean (SEM)	
Animal Group	Left Hind	Right Hind	Left Hind	Right Hind
Control (*n* = 6)	2.7 (2.0) ^a^	0.0 (1.3) ^a^	2.7 (1.2) ^a^	3.8 (0.8) ^a^
Inflamed (*n* = 6)	22.7 (10.6) ^a^	23.7 (10.5) ^a^	29.0 (9.0) ^b^	27.7 (7.4) ^b^
Ang-(1-7) (*n* = 5)	4.8 (1.3) ^a^	10.0 (10.7) ^a^	9.6 (2.5) ^ab^	12.4 (4.6) ^a^
Ang. Conj. (*n* = 5)	3.1 (1.5) ^a^	1.8 (0.6) ^a^	3.6 (1.6) ^a^	2.5 (1.4) ^a^

Values are reported as mean ± SEM, and statistical analysis was done using one-way ANOVA with the Tukey multiple comparison post-hoc test. In each column, groups labeled with different letters (a or b) have significant differences between them, *p* < 0.05.

**Table 2 pharmaceuticals-15-01157-t002:** List of the primer sequences used in a reverse transcription–quantitative polymerase chain reaction (qPCR) analysis of genes.

Gene	Primers	Sequences	Reference
ACE1	Forward (5′→3′)	TTTGCTACACAAATGGCACTTGT	[50]
	Reverse (5′→3′)	CGGGACGTGGCCATTATATT	
ACE2	Forward (5′→3′)	ACCCTTCTTACATCAGCCCTACTG	[51]
	Reverse (5′→3′)	TGTCCAAAACCTACCCCACATAT	
MasR	Forward (5′→3′)	AGAAATCCCTTCACGGTCTACA	[52]
	Reverse (5′→3′)	GTCACCGATAATGTCACGATTGT	
AT1R	Forward (5′→3′)	CCTCTACAGCATCATCTTTGTGG	[53]
	Reverse (5′→3′)	CACACTGGCGTAGAGGTTGA	
GAPDH	Forward (5′→3′)	CCTGCACCACCAACTGCTTA	[54]
	Reverse (5′→3′)	AGTGATGGCATGGACTGTGG	

## Data Availability

Data is contained within the article.

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
