# Peer review of "Anti-Inflammatory Effects of Ang-(1-7) Bone-Targeting Conjugate in an Adjuvant-Induced Arthritis Rat Model"

_pharmaceuticals, 2022, doi:10.3390/ph15091157_

Round 1

Reviewer 1 Report

In the present study the authors set out to investigate the potential therapeutic effects of Ang1-7 peptide conjugate in AIA. In the introduction they describe the role of the angiotensin system in the regulation of inflammation. Ang1-7 peptide is an anti-inflammatory mediator which has already been shown to have beneficial effect in antigen induced arthritis as well as adjuvant induced arthritis models. The main goal of the authors here to conjugate the Ang1-7 to a bone targeting component which would increase the otherwise very short half-life.

The main findings of the study include:

i) Ang1-7conj treatment decreased the clinical signs of arthritis slightly more than the peptide alone;

ii) Ang1-7conj decreased the NO levels in the serum;

iii) Higher serum Ang1-7 levels were measured in case of the Ang1-7conj treatment compared to the peptide alone conferring that the conjugation has enhanced the pharmacological properties;

iv) This achievement was confirmed on the molecular level as well: Ang1-7conj changed the expression of ACE1/2 and MasR, AT1R mRNA more effectively than the peptide alone, although this effect was less prominent at the protein level

Based on these, the authors conclude that the new Ang1-7conj could prove a useful new treatment modality in arthritis therapy.

Although the manuscript is well written, the experiments seem logical and the results are convincing, the reviewer has some concerns.

1. If the Ang1-7conj has really a longer life span than the peptide alone, a logical approach would be to change the treatment protocol, and decrease the number of treatments. Was there any such experiment?

2. It would also greatly validate the findings if the effect of the Ang1-7conj treatment would be analyzed on the histological level from the inflamed paws.

3. Added to the previous point, some of the immunological markers of arthritis could also be followed in the sera: eg. RF, anti-CCP, inflammatory cytokines (TNFa, IL-1b, IL-6, IL-17 etc).

4. A technical point: the annotation of significance in the figures is unclear: the letters (a,b,c etc) used for the labeling should be clearly defined in the figure captions.

Author Response

Comments and Suggestions for Authors

In the present study the authors set out to investigate the potential therapeutic effects of Ang1-7 peptide conjugate in AIA. In the introduction they describe the role of the angiotensin system in the regulation of inflammation. Ang1-7 peptide is an anti-inflammatory mediator which has already been shown to have beneficial effect in antigen induced arthritis as well as adjuvant induced arthritis models. The main goal of the authors here to conjugate the Ang1-7 to a bone targeting component which would increase the otherwise very short half-life.

The main findings of the study include:

  1. i) Ang1-7conj treatment decreased the clinical signs of arthritis slightly more than the peptide alone;
  2. ii) Ang1-7conj decreased the NO levels in the serum;

iii) Higher serum Ang1-7 levels were measured in case of the Ang1-7conj treatment compared to the peptide alone conferring that the conjugation has enhanced the pharmacological properties;

  1. iv) This achievement was confirmed on the molecular level as well: Ang1-7conj changed the expression of ACE1/2 and MasR, AT1R mRNA more effectively than the peptide alone, although this effect was less prominent at the protein level

Based on these, the authors conclude that the new Ang1-7conj could prove a useful new treatment modality in arthritis therapy.

Although the manuscript is well written, the experiments seem logical and the results are convincing, the reviewer has some concerns.

Answer: We appreciate the respected reviewer’s kind words emphasizing our study’s significant findings and providing us with constructive advice to improve its quality.

  1. If the Ang1-7conj has really a longer life span than the peptide alone, a logical approach would be to change the treatment protocol and decrease the number of treatments. Was there any such experiment?

Answer: Thank you for your valid suggestion. 

Using a gamma counter radioassay, we have shown that Ang. Conj. has a longer half-life [16]. The radioassay methods have some limitations in their application for the quantitative measurement of radiolabeled material. We are currently developing an LC-MS/MS method to quantify Ang. Conj. in plasma to confirm our previous PK results. Assuming a direct relationship between the plasma concentration of Ang-(1-7) and its anti-inflammatory effect, we investigated Ang. Conj. therapeutic effects using a rat model of RA compared to plain Ang-(1-7) and elucidated the impact of bone-targeting delivery and stability improvement strategy on enhancing their anti-inflammatory effects. To make the comparison more effective and the effect size more significant, we increased the dosing interval from the previously used daily regimen [22] to 3x/week.

We added this rationale to the "Introduction" section of the study.

  1. It would also greatly validate the findings if the effect of the Ang1-7conj treatment would be analyzed on the histological level from the inflamed paws.

Answer: Thanks for your constructive comment. According to the results of a study (Liu, J. et al.  Angiotensin-(1–7) attenuates collagen-induced arthritis via inhibiting oxidative stress in rats. Amino Acids 53, 171–181 (2021). https://doi.org/10.1007/s00726-020-02935-z), Ang-(1-7) improves paw histological markers changed by arthritis. For further investigation and to address the rheumatoid arthritis cardiovascular complications, in this study, we aimed to explore the cardiovascular effects of inflammation through RAS biomarkers and examine the protective effects of Ang Conj treatment. Similarly, this explanation has been added to the experimental sections.

  1. Added to the previous point, some of the immunological markers of arthritis could also be followed in the sera: eg. RF, anti-CCP, inflammatory cytokines (TNFa, IL-1b, IL-6, IL-17 etc).

Answer: We appreciate the reviewer’s point. In response to the previous comment, Ang-(1-7) treatment normalized the arthritis-induced cytokine levels (https://doi.org/10.1007/s00726-020-02935-z). Our study provides complementary data based on the RAS components at enzyme, peptide, and receptor levels in different vital tissues. We did include a justification for your suggestions in the manuscript.

  1. A technical point: the annotation of significance in the figures is unclear: the letters (a,b,c etc) used for the labeling should be clearly defined in the figure captions.

Answer: Thank you for your constructive suggestion. We did add the following explanation in the figures caption. “Data labeled with different letters (a, b, c, or d) indicate a statistical difference between groups where p < 0.05.”

Reviewer 2 Report

In this manuscript, the authors indicate that the novel and stable conjugate (Ang. Conj.) with an extended half-life of Ang-(1-7) provides therapeutic efficacy in a rat model of AIA. The manuscript shows some interesting points, however, for the given reasons the paper cannot be published as it stands.

Major points:

1.    Although the authors are comparing Ang and Ang. Conj. in an arthritis model, since bisphosphonate alone has also demonstrated anti-arthritic effects, the authors should make a simultaneous comparison in this study to demonstrate the enhanced effect of the longer blood half-life.

2.    Previous papers have reported the involvement of TNF alpha and other cytokines and their signaling in the anti-inflammatory effects of Ang. However, this point has not been discussed in this study. The authors should comprehensively consider the involvement of cytokines and signaling pathways involved with Ang and Ang. Conj.

3.    In addition, the authors should show an effect on bone destruction.

4.    In Fig.2B, the arthritis score seems very weak. The authors describe the following in the Methods section.  “An arthritis index score of ≥ 5 was considered infliction of the disease, which was evident typically in 8-12 days after adjuvant injection.” However, the onset of the disease is with arthritis of ≥ 5, but doesn't the administration start at that point? It should be clarified at what time the administration was started.

5.    I Fig.2B, the authors should discuss the fact that if the drug was started at 10 days, therapeutic efficacy was not observed until 20 days.

6.    In Fig.3, the author evaluated that the effect of Ang and Ang. Conj. on plasma NO concentrations are compared. On the other hand, the measurement of local NO concentrations in the joints may be more important for the therapeutic effect on arthritis.

7.    In Fig. 4, the author evaluated that the effect of Ang and Ang. Conj. on Ang concentration and Ang/AngII ratios are reversed to normal in Ang, but are significantly increased in the Ang. Conj. compared to normal. In this regard, the authors should consider the side effects of the combination therapy.

8.    Moreover, the author evaluated the plasma concentration of Ang recovered to the same level as in the normal group, but the anti-arthritis effect and plasma NO concentration did not recover normal levels. The authors should consider this point.

Miner points:

1.    In Fig. 1A, the authors should state at what body weight they are comparing.

2.    For all figures, it should be clearly stated to which group the significant difference alphabets refer.

3.    What do AA (Line 102) and AI (Line 110) indicate?

Author Response

In this manuscript, the authors indicate that the novel and stable conjugate (Ang. Conj.) with an extended half-life of Ang-(1-7) provides therapeutic efficacy in a rat model of AIA. The manuscript shows some interesting points, however, for the given reasons the paper cannot be published as it stands.

Answer: We are grateful that the reviewer found our manuscript findings interesting and provided constructive comments to help us improve the manuscript quality to be published in the prestigious journal of Pharmaceuticals.

Major points:

  1. Although the authors are comparing Ang and Ang. Conj. in an arthritis model, since bisphosphonate alone has also demonstrated anti-arthritic effects, the authors should make a simultaneous comparison in this study to demonstrate the enhanced effect of the longer blood half-life.

Answer: Thank you for the valid point. The bisphosphonate moiety used in the conjugate partially resembles the therapeutic bisphosphonates and does not have any significant pharmacological effect compared to the 1st and 3rd generations of the bisphosphonates. Others and we previously reported that thiol-BP has no significant pharmacological effects. ( Young et al.,  Drug Deliv. and Transl. Res. (2017) 7:482–496 DOI 10.1007/s13346-017-0407-2, https://doi.org/10.1016/j.biomaterials.2013.01.059). This explanation has been added to the manuscript.

  1. Previous papers have reported the involvement of TNF alpha and other cytokines and their signaling in the anti-inflammatory effects of Ang. However, this point has not been discussed in this study. The authors should comprehensively consider the involvement of cytokines and signaling pathways involved with Ang and Ang. Conj.

Answer: Thanks for your constructive comment. According to the results of a study (Liu, J., et al.  Angiotensin-(1–7) attenuates collagen-induced arthritis via inhibiting oxidative stress in rats. Amino Acids 53, 171–181 (2021). https://doi.org/10.1007/s00726-020-02935-z), Ang-(1-7) improves all cytokines and inflammatory biomarkers altered by arthritis. For further investigation and to address the rheumatoid arthritis cardiovascular complications, in this study, we aimed to explore the cardiovascular effects of inflammation through RAS biomarkers and examine the protective effects of Ang Conj treatment.  We appreciate the reviewer’s concern regarding cytokines and their anti-inflammatory effects on Ang Conj. But our study mainly focuses on how RAS becomes imbalanced in rheumatoid arthritis, and our Ang Conj helps mitigate inflammation through the RAS spectrum. This explanation has been added to the experimental sections, and we will include your suggestions in our upcoming papers.

  1. In addition, the authors should show an effect on bone destruction.

Answer: We appreciate the reviewer’s point. Similar to the response to the previous comment, Ang-(1-7) treatment Ang-(1-7) improves histological markers changed by arthritis. Our study provides complementary data based on the RAS components at enzyme, peptide, and receptor levels in different vital tissues. We will again include a justification for your suggestions in the manuscript.

  1. In Fig.2B, the arthritis score seems very weak. The authors describe the following in the Methods section. “An arthritis index score of ≥ 5 was considered infliction of the disease, which was evident typically in 8-12 days after adjuvant injection.” However, the onset of the disease is with arthritis of ≥ 5, but doesn’t the administration start at that point? It should be clarified at what time the administration was started.

Answer: Thank you for your comment. Based on reported results using this animal model ( Spencer ling, F. Jamali, Drug Metabolism, and Disposition April 2005, 33 (4) 579-586; DOI: https://doi.org/10.1124/dmd.104.002360), and other studies reported by our group, we noticed that pre-AIA  ( defined with AI less than 5) is a suitable model of inflammation for pharmacokinetic studies. Pre-AA is associated with little or no pain and discomfort compared to fully developed adjuvant arthritis. Similarly, this study followed the same approach to avoid unnecessary distress, while the inflammation effect was noticeable. We started drug treating animals when they presented with some sign of inflammation (at least involvement of one joint)  which usually happens around 8-10 days after inoculation with adjuvant. We did clarify and address the drug administration as suggested.      

  1. In Fig.2B, the authors should discuss the fact that if the drug was started at 10 days, therapeutic efficacy was not observed until 20 days.

Answer: Thank you for the helpful comment. We addressed the issue by discussing the onset of action in the text.

  1. In Fig.3, the author evaluated that the effect of Ang and Ang. Conj. on plasma NO concentrations are compared. On the other hand, the measurement of local NO concentrations in the joints may be more important for the therapeutic effect on arthritis.

Answer: Thank you for the valid comments. Please refer to our response to your comment # 2.

  1. In Fig. 4, the author evaluated that the effect of Ang and Ang. Conj. on Ang concentration and Ang/AngII ratios are reversed to normal in Ang, but are significantly increased in the Ang. Conj. compared to normal. In this regard, the authors should consider the side effects of the combination therapy.

Answer: Ang1-7 is an endogenous peptide, so it has no reported side effects even when it was used in very high concentrations to evaluate its toxic effect (Mordwinkin NM, et al. Toxicological and toxicokinetic analysis of angiotensin (1-7) in two species. J Pharm Sci. 2012 Jan;101(1):373-80. doi: 10.1002/jps.22730. Epub 2011 Aug 19. PMID: 21858825; PMCID: PMC3619381). The increased plasma concentration of Ang1-7 after treatment with Ang Conj doesn’t reach the high concentration observed in the toxicology study, which proved it safe. 

  1. Moreover, the author evaluated the plasma concentration of Ang recovered to the same level as in the normal group, but the anti-arthritis effect and plasma NO concentration did not recover normal levels. The authors should consider this point.

Answer: Thank you for your great observation and valid inference. We believe such variation in the results Ang peptide level and effect size (NO level) could be due to the small sample size. Although NO levels and anti-arthritic effects in figures 2 and 3 did not reach exactly back to the control level, the statistical analysis shows that there was no significant difference between the control and Ang. Conj treated rats.

Miner points:

  1. In Fig. 1A, the authors should state at what body weight they are comparing.

Answer: Body weight on day 24 is compared with day 0. This was added to the text.

  1. For all figures, it should be clearly stated to which group the significant difference alphabets refer.

Answer: Data labeled with different letters (a, b, c, or d) indicate a statistical difference between groups where p < 0.05.

  1. What do AA (Line 102) and AI (Line 110) indicate?

Answer: AIA is adjuvant-induced arthritis, and AI is arthritis index.

Reviewer 3 Report

First of all, it is a well-written manuscript with high quality of language and valuable experimental data. The study is well-designed using appropriate methods.

I only have some worries and remarks regarding the demonstration of the data and the description of the results:

1. In all of subheadings the authors should highlight the main result of the subsection. Please, use active sentences.

2. The figure legends are not uniform. In some cases, the clear titles of the legends (e.g., in case of Figure 2) and the precise description of the statistics are also missing (e.g., which type of ANOVA or which type of multiple comparisons test was used such as in case of Figure 2 and 3). The sample sizes and the letters showing significant differences on the graphs are also consequently not defined in either figure legends.

3. The authors should express the results as the mean ± S.E.M. instead of mean ± SD. If the goal is to compare means with ANOVA, the authors are more interested in showing how precisely the data define the mean than in showing the variability. Furthermore, the authors should add the F- and p-values for further strengthen the statistical significance of the results and calculate effect sizes using Cohen’s d (difference in means divided by pooled SD), which helps to the authors in drawing the appropriate conclusions.

4. Figure 2:

a.       The authors should describe the results according to the order of the panels.

b.       If the percentage changes of the values compared to the baseline were demonstrated on the graphs, the authors should describe the absolute values in parentheses in the text instead of repeating percentages which we can see in the figure.

c.       Please, shorten the titles of the y axis in panel A, C and D. “Change of body weight/joint diameter/paw diameter (%)” would be much more acceptable.

d.       Since the authors induced polyarthritis in the rats with a single tail root injection of the adjuvant, the demonstration of the means of both hind paws and joints would be acceptable instead of showing only the values of the right hind limb.

e.       The authors measured body weight, paw, and joint diameter thrice per week. Please, demonstrate the results using a line graph or clarify why only the results of one day is demonstrated and on which experimental day.

My formal comments are the following:

1. Line 78: 3-15 min, a space is missing.

2. Line 102, 110, 112 and 193: Please, define the abbreviations of “AA” and “AI”, and use them consequently. I assume these abbreviations are not the same as AIA.

3. Line 111: “In inflamed group” or “in inflamed rats”?

4. Line 121: Percentage change of body weight instead of “Body weight% change”

5. Line 130: neither

6. Line 135 and 148: Please, clarify what kind of level or concentration was measured. Plasma?

7. Line 156: but not the MasR expression?

8. Line 288: Please, clarify precisely on which experimental day the treatment started.

9. Line 299: Please, more precise. Quantification, Measurement or Determination of Serum NO Concentration?

10. Line 387-390: Please, revise this section according to the abovementioned suggestions.

Author Response

Comments and Suggestions for Authors

First of all, it is a well-written manuscript with high quality of language and valuable experimental data. The study is well-designed using appropriate methods. I only have some worries and remarks regarding the demonstration of the data and the description of the results:

Answer: We are very delighted with your kind words and positive points on the experimental design, language quality, and value of our findings. We highly value your constructive comments by addressing them accordingly.

  1. In all of subheadings the authors should highlight the main result of the subsection. Please, use active sentences

Answer: Thank you for your comment. As you recommended, we highlighted the main results under all subheadings.

  1. The figure legends are not uniform. In some cases, the clear titles of the legends (e.g., in case of Figure 2) and the precise description of the statistics are also missing (e.g., which type of ANOVA or which type of multiple comparisons test was used such as in case of Figure 2 and 3). The sample sizes and the letters showing significant differences on the graphs are also consequently not defined in either figure legends.

Answer: Thank you for your valid comments. All of your suggestions are implemented in the text accordingly.

  1. The authors should express the results as the mean ± SEM instead of mean ± SD. If the goal is to compare means with ANOVA, the authors are more interested in showing how precisely the data define the mean than in showing the variability. Furthermore, the authors should add the F- and p-values for further strengthen the statistical significance of the results and calculate effect sizes using Cohen’s d (difference in means divided by pooled SD), which helps to the authors in drawing the appropriate conclusions.

Answer: Thank you for your valid comments. All your suggestions are followed, and the required information is implemented in the text.

  1. Figure 2:
  2. The authors should describe the results according to the order of the panels.

Answer: Thank you for your suggestion. Results are now described in the same order as the panel.

  1. If the percentage changes of the values compared to the baseline were demonstrated on the graphs, the authors should describe the absolute values in parentheses in the text instead of repeating percentages which we can see in the figure.

Answer: Thank you for the comments. The absolute values are included. 

  1. Please, shorten the titles of the y axis in panel A, C and D. “Change of body weight/joint diameter/paw diameter (%)” would be much more acceptable.

Answer: Thank you for the suggestion. The y-axis legend is fixed and shortened.

  1. Since the authors induced polyarthritis in the rats with a single tail root injection of the adjuvant, the demonstration of the means of both hind paws and joints would be acceptable instead of showing only the values of the right hind limb.

Answer: Thank you for your comment. The means of both hind paws are reported.

  1. The authors measured body weight, paw, and joint diameter thrice per week. Please, demonstrate the results using a line graph or clarify why only the results of one day is demonstrated and on which experimental day.

Answer: Thank you for the suggestion. The change in weight and paw diameters are presented in line graphs. 

My formal comments are the following:

  1. Line 78: 3-15 min, a space is missing.

Fixed.

  1. Line 102, 110, 112 and 193: Please, define the abbreviations of “AA” and “AI”, and use them consequently. I assume these abbreviations are not the same as AIA.

Abbreviations are now included in the text.

  1. Line 111: “In inflamed group” or “in inflamed rats”?

The group’s name is now consistent over the manuscript.

  1. Line 121: Percentage change of body weight instead of “Body weight% change”

Fixed.

  1. Line 130: neither

Fixed.

  1. Line 135 and 148: Please, clarify what kind of level or concentration was measured. Plasma?

NO levels were measured in serum and clarified where ever is needed.

  1. Line 156: but not the MasR expression?

 It is fixed.

  1. Line 288: Please, clarify precisely on which experimental day the treatment started.

It has been clearly mentioned in the method section.

  1. Line 299: Please, more precise. Quantification, Measurement or Determination of Serum NO Concentration?

The experimental procedure was added to the text.

  1. Line 387-390: Please, revise this section according to the abovementioned suggestions.

It is fixed.

Round 2

Reviewer 1 Report

-

Reviewer 2 Report

Thank you for your detailed responses to individual comments.

The answers by authors are not sufficient to understand the content. The authors should also consider the possibility that the RAS system is influenced by cytokines. In particular, #2, #6 and #8 are considered very important in this paper.

In addition, authors should describe in the legend which group each significant difference mark was compared to in all figures.

Author Response

Comments and Suggestions for Authors

Thank you for your detailed responses to individual comments.

The answers by authors are not sufficient to understand the content. The authors should also consider the possibility that the RAS system is influenced by cytokines. In particular, #2, #6 and #8 are considered very important in this paper.

Answer: Thank you for your satisfaction with most of our answers to your comments. We elaborated more in detail about those three comments that you found our response insufficient. Please see below.

The response related to Comments # 2 and 6: 

Activating the RAS induces inflammation by signaling AT1R by Ang II on leukocytes. After RA's initiation and development, inflammatory cells infiltrate the articular synovial tissues and consequently secrete inflammatory cytokines, such as TNF-α, IL-1, and IL-6 [15]. In rodent models of AIA through augmentation of the RAS protective arm by Ang-(1–7) or the Mas agonist AVE 0991, the AIA-induced neutrophil accumulation and production of proinflammatory mediators such as TNF-a, IL-1β, and CXCL1 in prearticular tissue were diminished, and paw histopathological markers were significantly reduced [12]. Ang-(1-7) also improved paw histological changes and normalized cytokine biomarkers in the Collagen-induced arthritis rat model [16]. Furthermore, increased levels of NO in plasma [17,18], articular fluid [19]), and a high expression of iNOS in hyperplastic synovium[17,20] and chondrocytes [21] have been reported in different models of AIA. This elevation of NO and ROS are related to the presence of macrophages in the inflamed tissues due to higher expression of iNOS induced by inflammatory cytokines. In a study using cultured human aortic smooth muscle cells (HASMC), the expression of inducible nitric oxide synthase (iNOS) and the release of nitric oxide (NO) were stimulated by both Ang II and IL-1β. Treatment with Ang-(1-7) inhibits the NO production, which Mas receptor antagonists block such effect [22]. These findings indicate that Mas receptor activation by Ang-(1-7) prevents neutrophil influx, cytokine production, and No release and, as a result, significantly improves arthritis in rats and mice AIA models.  

 As mentioned above, Ang-(1-7) presented protective effects by reducing proinflammatory cytokine levels in the plasma, HASMC, and joint tissues. In the current study, we further investigated the impact of inflammation on the RAS at the cellular level and focused on exploring the cardiovascular complication of RA. We additionally studied the protective effects of Ang. Conj. treatment, as a stable form of Ang-(1-7), on the RAS in the enzyme, peptide, and receptor levels of the heart, kidney, liver, and lung tissues.

Increased plasma and synovial tissue NO levels due to high expression of iNOS have been reported in patients with RA [31,32]. In the rat model of AIA, systemic administration of selective and nonselective iNOS halted the development of the disease [17,33]. Similarly, we evaluated   Ang. Conj. systemic anti-inflammatory effects through assessment of serum concentration of NO.

The response related to Comment # 8: 

Although, the NO serum levels in Ang-(1-7) or Ang. Conj. treated groups did not reduce to exact the same level as the control group; there was no significant difference between their NO values. Similarly, the therapeutic efficacy of Ang-(1-7) and Ang. Conj. on reducing AI was not significantly noticeable until ten days after treatment started (Figures 2A and B). This observation could be attributed to the time required for Ang-(1-7) concentration to release from the conjugate and to reach an adequate steady-state concentration. The two weeks treatment period with a low dose of Ang-(1-7) or Ang. Conj. in this pilot study presents a promising positive outcome, which can be potentiated by increasing the dose and duration. These findings are mostly in line with reestablishing the disturbed balance between the classical and protective arms, which is presented as ratios of Ang-(1-7)/Ang II peptides (Figure 4), or ACE2/ACE1 and MasR/AT1R at the gene or protein levels (Figures 5 & 6).

In addition, authors should describe in the legend which group each significant difference mark was compared to in all figures.

Answer: Thank you for your comment. We performed a Tukey multiple comparison post-hoc test, and any mean value labeled with a unique letter indicates that it is significantly different from other groups. In other words, if two or three mean values carry the same letter, there is no significant difference between them. We modified the wording to "Groups labeled with different letters have significant differences between them (p<0.05)" to address your comment.

Reviewer 3 Report

Some points were still not acceded.

1. The letters showing significant differences between the groups are not defined in either figure legends. To which group did the authors compare in case of “a”, “b” and “c” etc.? Please explain in the legends (e.g. ap<0.05 vs. control group).

2. Figure 2: Probably we misunderstood each other. Since the percentage changes of the values compared to the baseline were demonstrated on the graphs, absolute values of the body weight and the paw and joint diameters in gram (g) and millimeter (mm) should be added to the text and/or the Table 1.

3. Line 158 and 170: Please include “The plasma level” and “The plasma concentration” instead of simply “level” and “concentration”, respectively.

4. Line 342: Please revise the subheading titled “Serum NO Concentration”. Measurement of the Serum NO Concentration would be much better.

Formal comment:

Table 1: Please define in the legend which type of ANOVA was used.

Author Response

Comments and Suggestions for Authors

Some points were still not acceded.

Thank you for your agreement with most of our responses. The following items are also addressed as you suggested.

  1. The letters showing significant differences between the groups are not defined in either figure legends. To which group did the authors compare in case of "a", "b" and "c" etc.? Please explain in the legends (e.g. ap<0.05 vs. control group).

Answer: Thank you for your comment. We performed a Tukey multiple comparison post-hoc test, and any mean value labeled with a unique letter indicates that it is significantly different from other groups. In other words, if two or three mean values carry the same letter, there is no significant difference between them. We modified the wording to "Groups labeled with different letters have significant differences between them (p<0.05)" to address your comment.

  1. Figure 2: Probably we misunderstood each other. Since the percentage changes of the values compared to the baseline were demonstrated on the graphs, absolute values of the body weight and the paw and joint diameters in gram (g) and millimeter (mm) should be added to the text and/or the Table 1.

Answer: Sorry for the misunderstanding; the absolute values are added to the text.

  1. Line 158 and 170: Please include "The plasma level" and "The plasma concentration" instead of simply "level" and "concentration", respectively.

It is fixed as suggested.

  1. Line 342: Please revise the subheading titled "Serum NO Concentration". Measurement of the Serum NO Concentration would be much better.

It is fixed as suggested.

Formal comment:

Table 1: Please define in the legend which type of ANOVA was used

It is fixed as suggested.

Round 3

Reviewer 2 Report

Thank you for your quick response. The manuscript has been properly revised. There are no more problems.

Reviewer 3 Report

The manuscript has been properly revised. I highly recommend the approval of the manuscript.